# A New Inert Natural Deep Eutectic Solvent (NADES) as a Reaction Medium for Food-Grade *Maillard*-Type Model Reactions

**DOI:** 10.3390/foods12091877

**Published:** 2023-05-02

**Authors:** Daniela Marianne Hartl, Oliver Frank, Corinna Dawid, Thomas Frank Hofmann

**Affiliations:** Food Chemistry and Molecular Sensory Science, TUM School of Life Sciences, Technical University of Munich, Lise-Meitner-Str. 34, 85354 Freising, Germany; daniela.hartl@tum.de (D.M.H.); corinna.dawid@tum.de (C.D.);

**Keywords:** natural deep eutectic solvents (NADES), food grade, *Maillard* model reactions, sucrose, D-sorbitol, UHPLC-MS, NMR, taste-modulating, nucleotide derivatives

## Abstract

Sustainability, low toxicity, and high solute potential are the fundamental reasons for focusing green chemistry on natural deep eutectic solvents (NADES). The application of NADES ranges from organic chemistry to the agricultural sector and the food industry. In the food industry, the desired food quality can be achieved by the extraction of small molecules, macromolecules, and even heavy metals. The compound yield in *Maillard*-type model reactions can also be increased using NADES. To extend the so-called “kitchen-type chemistry” field, an inert, food-grade NADES system based on sucrose/D-sorbitol was developed, characterized, and examined for its ability as a reaction medium by evaluating its temperature and pH stability. Reaction boundary conditions were determined at 100 °C for three hours with a pH range of 3.7–9.0. As proof of principle, two *Maillard*-type model reactions were implemented to generate the taste-modulating compounds *N*^2^-(1-carboxyethyl)guanosine 5′-monophosphate) (161.8 µmol/mmol) and *N*^2^-(furfuryl thiomethyl)guanosine 5′-monophosphate (95.7 µmol/g). Since the yields of both compounds are higher than their respective taste-modulating thresholds, the newly developed NADES is well-suited for these types of “kitchen-type chemistry” and, therefore, a potential solvent candidate for a wide range of applications in the food industry.

## 1. Introduction

In the past decade, green chemistry has focused on natural deep eutectic solvents (NADES) to achieve sustainability and simultaneous toxicity reduction [1,2,3]. Like the earlier concept of deep eutectic solvents (DES), by mixing two or more solid compounds such as sugars, sugar alcohols (also called polyols), organic acids, and quaternary ammonium compounds, NADES show a depression of the melting point compared to that of any of their individual components, which results in a liquid state of aggregation [2,3,4]. Dai et al. suggested the presence of hydrogen donors and acceptors and the resulting inter- and intramolecular hydrogen bonding as the reason for the formation of a liquid supramolecular structure from solid individual components. Nevertheless, ionic bonds and other physical intermolecular binding forces can also influence the structures of NADES and give them liquid ionic crystal-like properties [4]. Compared to DES, NADES ingredients are defined as primary metabolites or bio-based components. Therefore, NADES are cheap, non-toxic, environmentally friendly, edible, biodegradable, and sustainable liquids with versatile application possibilities depending on their components [2,3,4,5]. Dai et al. have discovered the most NADES systems in the literature [4], most of which correspond to NADES classes based on acid–base, polyol–acid, polyol–amino acid, polyol–base, sugar–acid, sugar–amino acid, and sugar–sugar combinations [6]. Water appears in NADES systems as strongly bonded water and cannot be evaporated. The stability, viscosity, and solubility capacity of the resulting NADES can be influenced by controlling the water content, preparation time, and temperature [2,4]. In addition to their roles as solvents, means of transport, and biocatalysts in living plant cells [2], NADES have also been used as extraction media and solvents for a wide range of polar and non-polar compounds [7], as well as small organic molecules such the aroma compound vanillin from vanilla pods [8], different colorful polyphenols from safflower [9], grape skin [10], and Flos Sophorae [11]; and macromolecules such as starch [4,12], gluten proteins [4,13], and DNA [4,5,14]. Huang et al. also showed that heavy metals such as cadmium can be extracted with yields of more than 99% with a choline chloride-based NADES system combined with saponin from rice to increase food quality [5,6,15]. In marine samples (fish muscle, liver tissues, and macroalgae), removing and determining Cu, Fe, Ni, and Zn was possible using a choline chloride–oxalic acid NADES [16,17]. Other applications of NADES reported in the literature to ensuring food safety and quality include the extraction and determination of ochratoxin A from wheat samples, breadcrumbs, and biscuits [17,18], as well as the extraction and determination of different pesticides from fruit juices and vegetables [17,19]. Furthermore, NADES can act as a stabilizing, activating, or catalytic reaction medium for enzymes such as lipases [5,20,21,22]. Kranz and Hofmann and Brehm et al. used NADES systems to increase the yield of taste-modulating molecules produced by so-called *Maillard* model reactions and opened new application possibilities for NADES in industrial use [23,24]. Generally, a *Maillard* reaction is defined as a reaction cascade initiated by the reaction of reducing sugars with amino compounds [25,26], which can lead to colored aroma- or taste-active molecules. For example, the umami taste-modulating compounds (*R*)- and (*S*)-*N*^2^-(1-carboxyethyl) guanosine 5′ monophosphate ((***R***)**-2**, (***S***)**-2**) were formed in yeast extracts by the reaction of guanosine 5′ monophosphate and the *Maillard* degradation product DL-glyceraldehyde [27]. In contrast, the umami-enhancing compound *N*^2^-(furfuryl thiomethyl) guanosine 5′-monophosphate (**1**) was formed by a *Maillard*-type model reaction of 2-furfurylthiol, formaldehyde, and guanosine 5′ monophosphate [28]. Previous studies have demonstrated that the ex-food production of (***R***)**-2** and (***S***)**-2** was increased to 215.3 µmol/mmol in a betaine/glycerol/water NADES system. The time required to generate these compounds was reduced from 10 days to 2 h using NADES and higher temperatures [23,27]. The *Maillard* compound 1-deoxy-D-fructosyl-*N*-*β*-alanyl-L-histidine, which leads to a white meat sensory perception, was also expanded from 12.2 µmol/mmol in aqueous buffered solution to 489.0 µmol/mmol in a glucose/sucrose/water NADES system (molar ratio of 1:1:9). Brehm et al. increased the yield of three different taste-modulating thiamine derivates [24,29] by using educts as a formulation of the NADES system for the first time. Therefore, NADES are a possible reaction media for the development of new savory taste-enhancing building blocks of food-grade quality [24] and open a field described in the literature as “next-generation culinary chemistry” [23]. The low water content of the system, as well as the fact that water is firmly bound in NADES [2,4], leads to low water activity. High water activity can generally lead to spoilage in food, such as microbial, enzymatic, physical, and chemical deterioration, which is why NADES with low water activity can also have preservative properties [22].

Consequently, NADES are a promising medium for the food industry to influence taste by simultaneously reducing table salt and monosodium glutamate and preserving food by lowering the water content and stabilizing molecules, e.g., polyphenols [22]. However, due to the legal limitations of many NADES ingredients, the food industry needs a NADES system with no regulation of any ingredients that are inert in order to avoid unpleasant side reactions.

Therefore, the aim of the current study is to develop a new food-grade NADES system that contains only ingredients with a low intrinsic reaction rate and nearly no concentration limits in foods. This NADES system will be extensively studied for temperature and pH stability to serve as a potential medium for *Maillard*-type model reactions. As proof of principle, the taste-modulating compound **1**, as well as (***R***)**-2** and (***S***)**-2** (structures: Figure 1), will be generated in the newly developed NADES system.

## 2. Materials and Methods

### 2.1. Chemicals

The following chemicals were obtained from commercial sources: sucrose (≥99.5%), D-sorbitol (≥98.0%), guanosine 5′-monophosphate disodium salt hydrate (5′-GMP) (≥99.0%), potassium phosphate monobasic (≥99.0%), sodium hydroxide (49.0–51.0%), formaldehyde (36.5–38.0% in water),^13^C, D_2_-formaldehyde (20.0% in D_2_O), and D/L-glyceraldehyde (≥90.0%), as well as the deuterated chemicals deuterium oxide (D_2_O), methanol-*d_4_*, dimethyl sulfoxide-*d*_6_ (DMSO-*d*_6_), and 3-(trimethylsilyl)propionic-2,2,3,3-*d*_4_ acid sodium salt (TMSP) from Sigma-Aldrich (Steinheim, Germany); sodium hydroxide (1 mol/L), sodium hydroxide (50.0–52.0%), hydrochloric acid (1 mol/L), and tyrosine (≥99.0%) from Fluka (Buchs, St. Gallen, Switzerland); formic acid (98.0–100.0%) and potassium hydroxide (≥99.0%) from Merck (Darmstadt, Germany); 2-furfurylthiol from TCI Germany GmbH (Eschborn, Germany); and caffeine (99.0%) from Alfa Aesar (Kandel, Germany). (***R***)**-2** and (***S***)**-2**, as well as their isotopically labeled internal standards ([^13^*C*_3_*]-*(*R*)*-N*^2^-(1-carboxyethyl)guanosine 5′-monophosphate ((***R***)**-2-IS**), [^13^*C*_3_]-(*S*)*-N*^2^-(1-carboxyethyl)guanosine 5′-monophosphate ((***S***)**-2-IS**), were synthesized as described by Festring and Hofmann [27,30].

For high-performance liquid chromatography (HPLC) and ultra-high-performance liquid chromatography—mass spectrometry (UHPLC-MS) measurements, water was purified using the Milli-Q reference A+ system from Merck Millipore (Darmstadt, Germany); methanol HPLC-grade, UHPLC-MS-grade, as well as acetonitrile HPLC-grade and UHPLC-MS-grade, were purchased from Thermo Fisher Scientific GmbH (Dreieich, Germany).

### 2.2. NADES Preparation

Two different methods from the literature were used to produce the NADES system, with some modifications: the heating and vacuum evaporation methods [4]. For the heating method, sucrose and D-sorbitol were weighed in equimolar ratios in a closed heat-stable centrifuge tube, and an eightfold molar ratio of water was added. The reaction mixture was heated at 70 °C between 120 min and 180 min under stirring until a clear liquid was obtained. For the evaporation method, sucrose and D-sorbitol were entirely dissolved in water and evaporated at 40 °C using a rotary evaporator (BÜCHI Labortechnik AG, Flawil, Switzerland) and dried in a desiccator until a constant weight was reached. The NADES systems were stored at 7 °C until further use.

### 2.3. Stability Tests of Suc/Sorb NADES under Different Heat and pH Conditions

The stability of the NADES ingredients sucrose and D-sorbitol was determined before and after heat or pH treatment by means of UHPLC-MS. In order to investigate its heat stability, NADES (1.0 g) was heated at different temperatures (40 °C, 70 °C, 90 °C, 100 °C, 110 °C, 120 °C, 130 °C, and 150 °C) for different time intervals from 0.5 h to 48 h. The pH stability tests were performed after adding a sodium hydroxide solution (10 µL, 0.1 mol/L) or hydrochloric acid (10 µL, 0.1 mol/L) to the NADES. For quantification, an aliquot (20 mg) of the sample was taken before dilution with water (5 mL), followed by pH determination using a Schott Lab 860 pH meter (SI analytics, Xylem Analytics Germany, Mainz, Germany). The aliquot was solved in 1.0 mL of a mixture of acetonitrile/water (20/80; *v*/*v*). After membrane filtration (0.45 µm) and dilution by a factor of 2000, 1 µL of the sample was injected into a Shimadzu Nexera X2 UHPLC system consisting of a controller unit (CBM-20A), two binary pumps (LC-30AD), an autosampler (SIL-30AC), a column oven (CTO-30A), and a degasser (DGU-20A5R) (Shimadzu Deutschland GmbH, Duisburg, Germany) coupled with a linear ion trap quadrupole (QTRAP) 6500 mass spectrometer system (AB Sciex Instruments, Darmstadt, Germany). The compounds of the NADES system were separated on an ACQUITY BEH Amid column (100 × 2.1 mm, 100 Å, 1.7 µm) with a corresponding security guard column (Waters GmbH, Eschborn, Germany) at a flow rate of 0.4 mL/min of a solvent mixture of water (solvent A) and acetonitrile (solvent B), each with the addition of 0.1% formic acid. The following gradient was used: starting at 20% A for 1.0 min; increasing to 50% A for 3.0 min; increasing at 100% A for 0.5 min; holding for 1.5 min at 100% A; decreasing at 20% A for 1.0 min; and holding for 2.5 min. For the 6500 mass spectrometer system, full scan mode was used in combination with multiple-reaction monitoring (MRM). Positive electrospray ionization mode (ESI^+^) and low-molecular-mass configuration were chosen. The ion spray voltage parameter was set to +5500 V, inert nebulizer gas was set to 55 psi, heater gas was set to 65 psi, curtain gas was set to 35 psi, entrance potential was set to +10 V, and ion source temperature was set to 450 °C. Declustering potential (DP), collision energy (CE), entrance potential (EP), and exit potential (CXP) were optimized for all fragments of the standard compounds sucrose (as sodium adduct; quantifier: *m/z* 365.009 🡪 203.000; qualifier: *m/z* 365.009 🡪 185.000) and D-sorbitol (quantifier: *m/z* 183.001 🡪 147.100; qualifier: *m/z* 183.001 🡪 129.000) by detecting the pseudomolecular ions [M + H]^+^ during constant flow injection of 7 µL/min into the system. For data acquisition and system monitoring, Analyst 1.6.3 (AB Sciex Instruments) was used, and data processing was carried out using MultiQuant 3.0.3 software (AB Sciex Instruments), as well as Microsoft Excel 2016 (Microsoft Corporation, Redmond, WA, USA) and OriginPro 2020 (OriginLab Corporation, Northampton, MA, USA). Quantitative values were calculated by external calibration in the range of 3 nmol/L–400,000 nmol/L and represented as the mean of five replicates and a minimum of two technical replicates with associated standard deviations. Accuracy was checked by spiking experiments with five different concentrations of both analytes (after diluting 750, 2000, 6000, 10,000, and 30,000 nmol/L) into the solvent mixture and determining recovery rates, as well as intraday precision (Appendix A).

### 2.4. Model Reactions and Isolation of ***1***

As a model reaction, the synthesis of **1** was carried out in a phosphate buffer or NADES system based on sucrose and D-sorbitol according to Suess et al. and Lu et al., with some modifications [28,31]. Therefore, formaldehyde (0.20 mmol, 15 µL of a 37% solution in water) or [*D*_2_,^13^*C*]-formaldehyde (0.20 mmol, 30 µL of a 20% solution in D_2_O) and 2-furfurylthiol (0.15 mmol, 15 µL) were added to NADES (1.0 g) or an aqueous phosphate-buffered solution (10 mL, 0.03 mol/L, pH = 7.0) for comparison of both solvents and heated for 4 h at 40 °C in a closed vessel in an aluminum block. After adding 5′-GMP (0.15 mmol, 61.1 mg), the reaction mixture was maintained at 40 °C for 16 h. Finally, only the NADES reaction mixtures were dissolved in water (10 mL), membrane-filtered (0.45 µm), and separated using reversed-phase high-pressure liquid chromatography coupled with an ultraviolet/visible light detector (RP-HPLC-UV/vis) (two pumps P 6.1L, detector MWD 2.1L, fraction collector: LABOCOL Vario-4000, software: PurityChrom Version 5.09.036 (Knauer Wissenschaftliche Geräte GmbH, Berlin, Germany)). Separation was achieved on a Luna pentafluorophenyl column (250 × 21.2 mm, 100 Å, 5.0 µm) with a corresponding guard column from Phenomenex (Aschaffenburg, Germany). The following gradient was used at a flow rate of 20 mL/min with a solvent mixture of acetonitrile (solvent B) and water (solvent A), each with formic acid (0.1%) added: starting with 0% B for 2.5 min; increasing within 16.0 min to 30% B; increasing within 3.5 min to 100% B; holding 100% B for 3.0 min. The effluent obtained at λ = 260 nm from the separation was divided into eight fractions. The solvent from each fraction was removed using a rotary evaporator, lyophilized (Martin Christ Gefriertrocknungsanlagen GmbH, Osterode, Germany), and stored at −20 °C until further use. Fraction five contained the targeted compound, which could be verified as **1** based on ultra-high-performance liquid chromatography–time-of-flight mass spectrometry (UHPLC-TOF-MS), UV/vis, and one- and two-dimensional nuclear magnetic resonance (NMR) spectroscopic data.

*N*^2^-(furfurylthiomethyl) guanosine 5′-monophosphate (**1**, Figure 1). UV/vis (water/acetonitrile, with 0.1% formic acid added, 70/30, *v*/*v*) λ_max_ = 260 nm. UHPLC-TOF-MS (ESI^−^) m/z 488.0656 ([M − H]^−^, measured); m/z 488.0647 ([M − H]^−^, calculated for C_16_H_19_N_5_O_9_PS^−^). UHPLC-TOF-MS (ESI^+^) m/z 490.0807 ([M + H]^+^, measured); m/z 490.0792 ([M + H]^+^, calculated for C_16_H_21_N_5_O_9_PS^+^). ^1^H-NMR (500.13 MHz, methanol-*d_4_*, 298 K, COSY) δ (ppm) 3.91 [s, 2 H, H-C(1″)], 4.12–4.18 [m, 1 H, H-C(5′)], 4.19–4.25 [m, 1 H, H-C(4′)], 4.36 [t, 1 H, *J* = 4.6 Hz, H-C(3′)], 4.58 [d, 1 H, *J* = 14.1 Hz, H-C(1‴_A_)], 4.62 [t, 1 H, *J* = 5.1 Hz, H-C(2′)], 4.68 [d, 1 H, *J* = 14.1 Hz, H-C(1‴_B_)], 5.98 [d, 1 H, *J* = 4.9 Hz, H-C(1′)], 6.24 [d, 1 H, *J* = 3.2 Hz, H-C(5″)], 6.31 [dd, 1 H, *J* = 2.0, 3.2 Hz, H-C(4″)], 7.40 [dd, 1 H, *J* = 0.8, 1.9 Hz, H-C(3″)], 8.35 [s, 1 H, H-C(8)]. ^13^C-NMR (125 MHz, methanol-*d_4_*, 298 K, HMBC, HSQC) δ (ppm) 28.4 [CH_2_, C(1″)], 44.4 [CH_2_, C(1‴)], 66.4 [d, CH_2_, *^2^J_C,P_* = 4.7 Hz, C(5′)], 71.7 [CH, C(3′)], 75.9 [CH_2_, C(2′)], 85.1 [d, CH,*^2^J_C,P_* = 7.8 Hz C(4′)], 89.9 [CH, C(1′)], 108.5 [CH, C(4″)], 111.5 [CH, C(5″)], 115.8 [C, C(5)], 137.7 [CH, C(8)], 143.5 [CH, C(3″)], 151.9 [C, C(4)], 153.4 [C, C(2″)], 153.9 [C, C(2)], 158.0 [C, C(6)].

All data obtained for **1** aligned with the literature [28]. The threefold isotopically labeled analog, *[D*_2_, ^13^*C]-N^2^-*(furfurylthiomethyl) guanosine 5′-monophosphate (**1-IS**), was isolated in the same way.

[*D*_2_, ^13^*C*]-*N*^2^-(furfurylthiomethyl) guanosine 5′-monophosphate (**1-IS,**
Figure 1). (Water/acetonitrile, with 0.1% formic acid added, 70/30, *v*/*v*) λ_max_ = 260 nm. UHPLC-TOF-MS (ESI^−^) m/z 491.0800 ([M − H]^−^, measured); m/z 491,0806 ([M − H]^−^, calculated for *[*^2^*H*_2_*,* ^13^*C]*-C_16_H_19_N_5_O_9_PS^−^). UHPLC-TOF-MS (ESI^+^) m/z 493.0956 ([M + H]^+^, measured); m/z 493.0951 ([M + H]^+^, calculated for *[^2^H_2_, ^13^C]*-C_16_H_21_N_5_O_9_PS^+^). ^1^H-NMR (600 MHz, methanol-*d_4_*, 298 K, COSY) δ (ppm) 3.91 [d, 2 H, *J* = 4.6 Hz, H-C(1″)], 4.08–4.21 [m, 1 H, H-C(5′)], 4.23–4.29 [m, 1 H, H-C(5′)], 4.40 [t, 1 H, *J* = 4.4 Hz, H-C(3′)], 4.66 [t, 1 H, *J* = 5.0 Hz, H-C(2′)], 6.00 [d, 1 H, *J* = 5.0 Hz, H-C(1′)], 6.26 [d, 1 H, *J* = 3.1 Hz, H-C(5″)], 6.33 [dd, 1 H, *J* = 1.8, 3.3 Hz, H-C(4″)], 7.41 [dd, 1 H, *J* = 0.7, 1.9 Hz, H-C(3″)], 8.42 [s, 1 H, H-C(8)]. ^13^C-NMR (125 MHz, methanol-*d_4_*, 298 K, HMBC, HSQC) δ (ppm) 28.2 [CH_2_, C(1″)], 43.3–44.5 [m, CH_2_, C(1‴)], 65.9 [d, CH_2_, *^2^J_C,P_* = 4.1 Hz, C(5′)], 71.6 [CH, C(3′)], 75.7 [CH_2_, C(2′)], 85.2 [d, CH, *^2^J_C,P_* = 8.7 Hz C(4′)], 89.4 [CH, C(1′)], 108.6 [CH, C(4″)], 111.6 [CH, C(5″)], 115.3 [C, C(5)], 138.0 [CH, C(8)], 143.5 [CH, C(3″)], 152.0 [C, C(4)], 153.0[C, C(2″)], 153.7 [C, C(2)], 158.3 [C, C(6)].

### 2.5. Food-Grade Model Reaction of (**R**)**-*2***, (**S**)**-*2***

The formation of (***R***)**-2** and (***S***)**-2** in Suc/Sorb NADES was carried out according to the literature [23]. In NADES (8 g), a binary mixture of D,L-glyceraldehyde (0.33 mmol) and 5′-GMP (0.33 mmol) was dissolved and heated at 100 °C for 2 h. After 2 h, the reaction mixture was supplemented with water to a final volume of 100 mL [23].

### 2.6. Determination of the Exact Mass and Mass Fragmentation of ***1*** and Its Isotopically Labeled Analog **1-IS** Using UHPLC-TOF-MS

High-resolution mass spectra of aliquots of all fractions and the isolated compound **1** and its isotopically labeled analog (**1-IS**, 0.1 mg/mL) were generated using a Synapt G2-S high-definition mass spectrometer (HDMS) (Waters) combined with an Acquity UPLC core system (Waters). A 150 × 2.1 mm, 130 Å, 1.7 µm BEH C18 column with a corresponding guard column (Waters) was used for chromatic separation. A mixture of water (solvent A) and acetonitrile (solvent B), each with an additive of 0.1% formic acid and a gradient starting with 5% B, increasing within 4.0 min to 100% B, holding for 0.5 min, and decreasing within 0.5 min to 5% B, was used. All instrument parameters were applied according to [32]. Data acquisition and processing were carried out using MassLynx 4.1 SCN 8.5.1 (Waters).

### 2.7. NMR Spectroscopy

All one- (^1^H, ^13^C) and two-dimensional NMR measurements (^1^H, ^1^H correlation spectroscopy (COSY); ^1^H, ^13^C heteronuclear single-quantum coherence (HSQC); ^1^H, ^13^C heteronuclear multiple-bond correlation (HMBC); ^1^H, ^13^C heteronuclear single-quantum coherence rotating frame Overhauser enhancement spectroscopy (HSQC-ROESY)) for structural verification, as well as NADES characterization experiments, were conducted on an Avance NEO 500 MHz system equipped with a cryoprobe (CP 2.1 TCI, 300 K) or on a 600 MHz Avance NEO spectrometer equipped with a cryoprobe TCI at 300 K (for HSQC-ROESY 323 K). Data acquisition and analysis were performed using TopSpin 4.1.1 software (Bruker, Rheinstetten, Germany).

Quantitative NMR measurements (qHNMR) were performed on a Bruker AVANCE III 400 MHz spectrometer equipped with a Z-gradient 5 mm BBI probe (Bruker). Therefore, reference compounds (1–2 mg) were weighed in NMR tubes (178 mm × 5 mm, Z172600 USC tubes, Bruker, Faellanden, Switzerland), dissolved in methanol-*d*_4_ (600 µL)*,* and measured after spectrometer calibration using the external calibration standards of caffeine (3.58 mmol/L) and tyrosine (4.34 mmol/L) in D_2_O. After phase and baseline correction, as well as signal integration, the ERETIC II function (Electronic REference To access In vivo concentrations), which is based on the PULCON (PULse length-based CONcentration determination) methodology of TopSpin 3.6 (Bruker), was used for calculation of the concentration of each analyte solution [33].

For the NADES characterization experiments, NADES samples (0.15 g to 1.20 g) were diluted with D_2_O (20–90% (*w*/*w*)) after adding 150 µL of a solution of TMSP in D_2_O (12.0 mmol/L). For the HSQC-ROESY experiments, coaxial system complete insert tubes (2.97 mm; SP Wilmad-LabGlass, Vineland, NJ, USA) were assembled with DMSO-*d*_6_ containing 0.03% (*v*/*v*) TMS in the outer tube and pure NADES in the inner tube. TopSpin 3.6 and 4.0.9 software (Bruker) and MestReNova 14.2.2 (Mestrelab Research, Santiago de Compostela, Spain) were used for data processing.

### 2.8. Quantification of ***1***, (**R**)**-*2***, and (**S**)**-*2*** via UHPLC-MS

Identification and quantification of all synthesized compounds (**1**, (***R***)**-2,** (***S***)**-2**) in NADES and phosphate-buffered systems for comparison were performed using UHPLC-MS analysis. Reaction mixtures were prepared to generate compound **1** or food-grade formations of substances (***R***)**-2** and (***S***)**-2** were membrane-filtered (0.45 µm) and diluted (**1**: factor of 10,000 (quantifier: *m/z* 490.151 🡪 375.900; qualifier: *m/z* 490.151 🡪 164.000); (***R***)**-2** (quantifier: *m/z* 436.231 🡪 224.000; qualifier: *m/z* 436.231 🡪 178.000) and (***S***)**-2** (quantifier: *m/z* 436.172🡪 224.100; qualifier: *m/z* 436.172 🡪 178.100): factor of 1000). After adding an internal standard (100 µL of **1-IS** (14.1 µmol/L) (quantifier: *m/z* 493.232🡪 167.100; qualifier: *m/z* 493.232 🡪 135.100) or 40 µL of (***R***)**-2-IS** (quantifier: *m/z* 439.250🡪 227.000; qualifier: *m/z* 439.250 🡪 180.000) or (***S***)**-2-IS** (5.0 µmol/L) (quantifier: *m/z* 439.073🡪 135.000; qualifier: *m/z* 439.073 🡪 163.000)) and mixing, an aliquot of 1 µL was analyzed by UHPLC-MS/MS using a 6500 mass spectrometer (AB Sciex Instruments) (instrument parameters c.f. 2.3). Chromatography was performed on a Kinetex pentafluorophenyl column (100 × 2.1 mm, 100 Å, 1.7 µm, Phenomenex) with a flow rate of 0.4 mL/min. The solvent mixture consisted of water (solvent A) and methanol (solvent B), each with an addition of 0.1% formic acid, and the following gradient was used for separation: starting at 100% A for 1.0 min; increasing B to 50% for 0.5 min; increasing at 60% B for 2.0 min; increasing at 100% B for 0.5 min; holding for 1.5 min at 100% B. For quantification, a combination of internal and external calibration in the range of 1.86 nmol/L–13647 nmol/L was used and represented as the mean of three replicates with associated standard deviations. (***R***)**-2** and (***S***)**-2** were referenced to the molar content of the educt D,L-glyceraldehyde. Statistical values (*t*-test for independent samples) were calculated by Microsoft Excel 2016 (Microsoft Corporation). All structures, including the internal standards, are presented in Figure 1, and all quantitative values, including standard deviations, are presented in Appendix A.

## 3. Results and Discussion

### 3.1. Development and Characterization of a New NADES System

Recent investigations revealed that NADES systems can conquer the yield limitations of *Maillard* model reactions and open a new field of “food-grade kitchen-type chemistry” [23]. Therefore, a new NADES system suitable for food should be developed that contains only ingredients with a low reaction rate and nearly no concentration limits. Many NADES are made of quaternary ammonium compounds such as choline or betaine, but both compounds are currently not allowed in foods in the European Union (EU) [34]. Therefore, the individual NADES ingredients must be part of the EU’s positive list of food additives and have a low reaction rate, functioning only as a solvent and not as an educt. Therefore, mainly natural primary metabolites with a low reaction rate were chosen as NADES ingredients, such as non-reductive sugar sucrose and sugar alcohol (also called polyol) D-sorbitol (Suc/Sorb NADES). For D-sorbitol, a laxative threshold of 71 g/day has been established [35]. Therefore, a recommended maximum dose of 50 g/day may not be exceeded, although D-sorbitol is generally rated as safe [34,36,37,38,39].

Using the evaporation method [4], the optimum molar ratio of both NADES ingredients was investigated. The successful formation of a NADES system was defined by creating a viscous liquid after evaporation at room temperature without solid particles [4]. For the Suc/Sorb NADES, which can be classified as sugar–polyol NADES, it was even possible to build a stable NADES system at a molar ratio of 1:1. To determine the minimal molar water content required to form a stabilized NADES system, a threefold approach was weighed and dried in a desiccator for 10 days at room temperature. Recrystallization started between the seventh and tenth days; therefore, the stability of the NADES system can be guaranteed only until the seventh day at room temperature. The evaporation method is suitable for evaluating the molar ratios of the individual components relative to one another, but for industrial applications, a system with fixed water content is required. Therefore, the NADES system was prepared and evaluated with different molar water ratios (1:1:4, 1:1:5, 1:1:6, 1:1:7, and 1:1:8 Suc:Sorb:water, respectively) using the heating method [4]. Finally, a transparent liquid was obtained from a fixed molar ratio of 1:1:8 (Appendix A). At a water content lower than the molar ratio of eight, crystals remained in the solution using the heating method. The observation that different proportions of NADES ingredients or water affect stability and physical properties such as viscosity or chemical properties such as solubility capacity of NADES is known in the literature [4].

Physicochemical properties are influenced by inter- and intramolecular interactions, which are expected to be hydrogen bondings of water and the hydroxy groups of both NADES ingredients. Sugars and sugar alcohols can function as hydrogen-bond donors and acceptors [4,40]. NMR spectroscopy experiments are widely used as an analytical technique for objectively evaluation of non-covalent interactions [40]. The often-used NOESY [2,4] or HOESY [4] experiments for determining intermolecular interactions are limited by the high viscosity of NADES and, thus, the separation of the different sugar and sugar alcohol signals [41]. Liu et al. recommended avoiding the high viscosity of NADES during NMR measurements by temperature regulation or dilution effects [41]. Therefore, NMR titration experiments from 20 to 90% (*w*/*w*) with D_2_O were performed and analyzed by ^1^H-NMR. All observed signals of both NADES ingredients and the residual proton signal of water showed chemical shifts to higher frequencies by increased dilution, including a decrease in signal broadening and an increase in intensity at higher dilution levels. Differences in chemical shifts of two exemplary signals (H-C(3′) and H-C(4′) of sucrose) determined precisely in Hertz are shown in Figure 2.

The chemical shifts of the individual signals were measured at each dilution, and the difference in the chemical shift was calculated in Hertz. The following data refer to the difference between two consecutive dilutions, e.g., H-ÅC(3′) showed a chemical shift difference of 2.7 Hz at dilutions of 50 and 60%. A complete signal assignment was performed by comparing reference spectra of the pure compounds (sucrose and D-sorbitol) with the spectra of the NADES system.

The highest chemical shift difference in the NADES system was observed for sucrose signals H-C(4′) and the anomeric proton H-C(1) from 20 to 30% of 5.9 Hz or 5.6 Hz, respectively. All signals were shifted to lower frequencies at low dilution levels of the NADES. In contrast, an increasing dilution ratio leads to a deshielding effect of the observed protons to higher frequencies. Beyond a dilution level of 40 to 50%, the chemical shift differences are reduced. For example, for sucrose H-C(3′) from 70 to 80%, the chemical shift difference was only 1.8 Hz (Figure 2). For D-sorbitol signals, the same trend was observed. Strong intermolecular interactions (e.g., hydrogen bonds and dipole–dipole interactions) are expected at lower dilutions, resulting in a higher electron density in the environment of the observed proton and, therefore, stronger shielding and a shift to lower frequencies. With increasing dilution, the supramolecular NADES structure becomes weaker. This is reflected by the fact that the observed protons showed a deshielding effect to higher frequencies and a shorter signal width at half height in the ^1^H NMR spectrum. This observation is in line with the literature [42,43,44]. For example, Dai et al. found chemical shift differences by analyzing ^1^H-NMR spectra of 0–100% (*v*/*v*) diluted 1,2-propanediol–choline chloride–water NADES for most of the signals of both components within increasing dilution levels to higher frequencies. At a dilution level of 50% or higher, complete rupture of the hydrogen bonds was observed [42]. It must be considered that most ^1^H-NMR-based studies have focused on choline chloride NADES systems [42,43,44,45], wherein, besides hydrogen bondings, other intermolecular interactions, such as ionic interactions, were included and found to be capable of influencing the chemical shifts [40,45,46].

In contrast to the typical ionic NADES, all observed chemical shifts to higher frequencies of all signals indicate the presence of hydrogen bonds/dipole–dipole interactions in the Suc/Sorb NADES made of neutral compounds [42,43,44,45].

### 3.2. Thermal Stability of Suc/Sorb NADES System

It is of substantial interest to have a system with sufficient thermal stability to use the NADES as a food-grade solvent to generate *Maillard* reaction products. First, the Suc/Sorb NADES system was heated in 30 min intervals over a temperature range from 50 °C to 150 °C. The system’s temperature was increased by 20 °C every 30 min. At 150 °C, the total heating time was set to 90 min with sampling in 30 min intervals. To evaluate the stability of the Suc/Sorb NADES, both NADES ingredients were quantified using UHPLC-MS/MS (Figure 3a). Validation of the UHPLC-MS/MS method for external quantification of sucrose and D-sorbitol was conducted with five different recovery rates ranging from 80.6% to 118.8%, with intraday precision of 3.8% (D-sorbitol) and 5.7% (sucrose) (Appendix A).

The chemical stability of the Suc/Sorb NADES system was defined as an unchanged concentration or chemical degradation of the ingredients after heat treatment. While the physical properties of all samples were consistent during heat treatment, the quantitative data for sucrose showed a sharp decrease in the content up to the complete degradation of the compound at 150 °C. At 130 °C, 78.4% of the initial sucrose concentration could be detected. In contrast, the yields of D-sorbitol remained nearly constant over the entire temperature range. Only after 90 min at 150 °C was D-sorbitol degraded to approx. 82% of its initial concentration. To ensure a stable Suc/Sorb NADES system, the maximum temperature should be no higher than 130 °C with a heating duration of a maximum of 30 min. For heating intervals longer than 30 min, in a second experimental setup, the NADES system was heated for three hours at 120 °C and 100 °C (Figure 3b).

Neither at 100 °C nor at 120 °C could a degradation of D-sorbitol over three hours be observed. According to the literature, D-sorbitol exhibits inert properties at elevated temperatures. It is mainly resistant to amines, which react with reducing sugars to so-called *Amadori* compounds, with a subsequent complex reaction cascade, i.e., the *Maillard* reaction pathway [39]. Compared to D-sorbitol, the sucrose concentration decreased to 36.9% of the initial concentration over three hours at 120 °C. In contrast, at 100 °C, the initial and final sucrose concentrations differed by only 3.4% after three hours. Therefore, the reaction conditions for the use of the Suc/Sorb NADES system as a solvent for *Maillard*-type kitchen-like model reactions were set at 100 °C for three hours or 130 °C for 30 min.

In some cases, it might be necessary to avoid high temperatures to prevent further reactions of the desired aroma and taste-active or taste-modulating reaction products [47] produced in the NADES system. According to the Code of Practice of the International Organization of the Flavor Industry, a temperature decrease of 10 °C results in a doubling of the heating time; therefore, longer heating times can be selected at lower temperatures for the NADES system [48]. For subsequent reactions, e.g., for the production of compound **1** (Figure 1), a reaction time of 24 h at 40 °C is required. For this reason, reaction conditions were checked at this temperature/time combination, and no changes in the concentration ratio of the individual components could be observed (Appendix A). Therefore, the Suc/Sorb NADES system can also be used for long-term *Maillard* model reactions under moderate conditions of 40 °C.

### 3.3. pH Stability of Suc/Sorb NADES System

In addition to thermal stability, pH stability is also essential for food-grade systems. Previous experiments were performed with a constant pH value of the NADES system. The natural pH value of the Suc/Sorb NADES system was determined to be 5.7 because sugars in water show a slightly acidic pH value [49]. A defined amount of hydrochloric acid or sodium hydroxide solution was added after NADES formation to investigate the influence of the pH value. The pH value of all NADES samples was measured after dilution with 5 mL of water. Quantifying sucrose at different pH values at room temperature showed that sucrose degradation could be observed at very low pH values, such as pH 2.7. Both ingredients are stable at room temperature under slightly acidic (higher than pH 3.0), neutral, or alkaline conditions up to pH 9.0 (Figure 4a).

Appendix A shows the concentrations of both NADES ingredients at different pH values (3.7, 5.5, 9.0) after heat treatment at 40 °C for 48 h. Sucrose showed a slight decrease in concentration to 74.4% of the initial value at pH 3.7, whereas, for D-sorbitol, no changes in the concentration under different pH values were observable. This aligns with the literature, which reports that D-sorbitol shows high stability under acidic and alkaline conditions [39]. Therefore, the focus will be on sucrose for further investigations of the pH value. Figure 4 shows the pH shift of three Suc/Sorb NADES samples (pH 3.7, 5.7, and 9.0) before (Figure 4a) and after heat treatment at 100 °C for three hours (Figure 4b). The pH of the acid-treated sample dropped from 3.7 to 3.4, and the initial sucrose concentration was completely degraded. The concentration of sucrose under alkaline conditions (pH 9.0, as well as at the natural pH value (pH 5.7) of the NADES system, was relatively stable under heat treatment at 100 °C for three hours. This pH range of 5.7–9.0 corresponds to the literature ranges of pH 6.5–8.5 for minimum sucrose degradation at 100 °C for eight hours [50]. At lower pH values, acidic hydrolysis of the glycosidic bond and further sugar degradation of the resulting D-glucose and D-fructose moieties is likely to occur [51]. Generally, all samples showed a drop in the pH value after heat treatment, which became more evident at higher heating temperatures.

In summary, for the selected Suc/Sorb NADES, sucrose is the limiting factor; for this reason, the newly developed NADES system can be used as a solvent for *Maillard*-type model reactions with heat treatment up to 100 °C for three hours in a pH range of 5.7–9.0 and without heat treatment in the pH range of 3.7–9.0.

### 3.4. Formation of ***1*** in Suc/Sorb NADES

As an application for *Maillard*-type model reactions in Suc/Sorb NADES, the formation of the taste-modulating compound **1** was implemented with some modifications according to the literature [28]. The reaction of the educts 2-furfurylthiol, 5′-GMP, and formaldehyde carried out in phosphate buffer solution was compared to the new reaction medium on a quantitative level using stable isotope dilution analysis (SIDA) [52,53]. In Suc/Sorb NADES, a concentration of 95.7 µmol/g for **1** was determined, which is more than nine times higher than the concentration of the aqueous phosphate-buffered system of 10.4 µmol/g of **1**. Therefore, the new system can be considered suitable for the *Maillard*-type formation of **1** in high yields.

Compound **1** shows a 3.1 times higher taste-modulating effect than inosine 5′ monophosphate and a 1.3 times higher taste-modulating effect than 5′-GMP [28]. For comparison, the modulating taste threshold in monosodium glutamate solution for inosine 5′ monophosphate is approx. 5.2 µmol/L [54]. The NADES system will be consumed in diluted form for typical food applications. Therefore, a recalculation in 10 mL of water resulted in a concentration of 9.5 µmol/L of **1**. Therefore, the implemented reaction in NADES yielded around two times the modulating threshold concentration. Moreover, the low water activity and the high sugar content of the NADES system can show preservative properties [22]. Therefore, it has an application advantage compared to the buffer system in the food industry.

### 3.5. Formation of (**R**)**-*2*** and (**S**)**-*2*** as a Food-Grade Model Reaction in NADES

The food-grade reactions in NADES systems from the literature are the formation of the taste enhancers *N*-(1-methyl-4-oxoimidazolidin-2-ylidene) aminopropionic acid, 1-deoxy-D-fructosyl-*N*-*β*-alanyl-L-histidine, and (***R***)**-2** and (***S***)**-2** (Figure 4) [23]. As an application for *Maillard*-type model reactions, (***R***)**-2** and (***S***)**-2** were produced in the new Suc/Sorb NADES, as well as for comparison in sucrose/d-glucose (Suc/Glc) and betaine/glycerol (betaine/GlyOH) NADES systems using an equimolar mixture of 5′-GMP and D,L-glyceraldehyde and the same parameter according to the literature [23]. Quantitation of the compounds was performed by SIDA [30,52,53] using the respectively labeled compounds, as displayed in Figure 4**.** Besides the quantitated results of (***R***)**-2** and (***S***)**-2** in all tested NADES systems, the contents of the ex-food production from the literature as the sum of (***R***)**-2** and (***S***)**-2** in betaine/GlyOH, Suc/Glc, malic acid/sucrose- (Malic/Suc), and choline chloride/sucrose-NADES system (ChCl/Suc) are shown in Figure 5 [23].

The resulting yield of (***R***)**-2** and (***S***)**-2** (sum of both isomers: 161.8 µmol/mmol) in the Suc/Sorb NADES system was in the same concentration range as that of ex-food production in Suc/Glc (sum: 167.0 µmol/mmol). In betaine/GlyOH, which was established in the literature as the NADES system with the highest ex-food production of (***R***)**-2** and (***S***)**-2**, the sum of both isomers was 215.3 µmol/mmol [23]. The quantified content of (***R***)**-2** and (***S***)**-2** of 80.6 µmol/mmol in betaine/GlyOH differs from the literature value by 134.7 µmol/mmol. The determined yields of (***R***)**-2** and (***S***)**-2** in Suc/Sorb and Suc/Glc NADES systems exhibit a significant difference (*α* = 0.0001) compared to the yield achieved by betaine/GlyOH NADES. The entire determined content of 167.0 µmol/mmol in Suc/Glc differs from the published content in Suc/Glc (104.8 µmol/mmol) [23] by 62.2 µmol/mmol. The published concentrations of Malic/Suc and ChCl/Suc [23] were lower than those of betaine/GlyOH and Suc/Glc. The two NADES systems with the highest yields for (***R***)**-2** and (***S***)**-2** in the literature (Suc/Glc and betaine/GlyOH) showed comparable or lower values relative to the new Suc/Sorb NADES system.

Nevertheless, it must be taken into account that due to the complex network of different reactions and the numerous influencing factors, such as temperature, time, the composition of the system, water activity, and the pH value [25,55,56], the *Maillard* reaction is difficult to compare and to control [26,56]. For example, the *Maillard* reaction’s kinetics show high variability of reported activation energies, even for the same systems [56].

For the food industry, it is essential to have a system with a limited number of side reactions. Since D-glucose, as a reducing sugar, undergoes the *Maillard* reaction with suitable amino components, Suc/Glc NADES is unsuitable as a pure solvent for *Maillard* model reactions. In addition, possible side reactions cannot be avoided with betaine due to the functional acid group. In addition to its reactivity, betaine is currently not allowed in foods in the EU [31]. Therefore, the betaine/GlyOH NADES system is also inadequate for food applications.

The formation ratio of the diastereomers (***S***)**-2** (90.4 µmol/mmol) and (***R***)**-2** (71.4 µmol/mmol) in the Suc/Sorb NADE system must be considered, since (***S***)**-2** is a significantly more active taste enhancer than (***R***)**-2** [57]. The percentage formation of the diastereomer (***S***)**-2** was in a similar range of 54.5% (Suc/Glc), 55.9% (Suc/Sorb), and 60.1% (betaine/GlyOH) in the three tested NADES systems. This result is in contrast to the data published by Kranz and Hofmann showing that (***R***)**-2** is favored (between 65.0% and 80.0%) by performing the reaction in betaine/GlyOH NADES [23]. The used educt, D,L-glyceraldehyde, may influence the distribution of (***R***)**-2** and (***S***)**-2** [23]. Festring and Hofmann proposed a reaction pathway of (***R***)**-2** and (***S***)**-2** via methylglyoxal as a 3-deoxyoson. Due to the oxidation of the OH-group at C2 of D,L-glyceraldehyde to a keto function (methylglyoxal), the chirality is lost. As a result, a mixture of both diastereomers was obtained [57]. A tendency of slightly increased formation of (***S***)**-2** relative to (***R***)**-2** is also evident, according to Festring and Hofmann [27].

Festring and Hofmann determined an intrinsic umami taste threshold of 0.19 mmol/L for (***S***)**-2** and 0.85 mmol/L for (***R***)**-2** [27]. Sensory investigations of the taste-modulating effect of these two compounds showed an umami-enhancing effect for (***R***)**-2**, with a *β* value of 0.08 and of 7.0 for (***S***)**-2** related to the umami taste activity of inosine 5′-monophosphate [57]. This *β* value represents the potency of the sensory difference of the tested 5′-GMP derivate in comparison to 5′-IMP, each dissolved in sodium glutamate solution [51,56]. For typical food application, the concentration of the more taste-modulating (***S***)**-2** isomer in Suc/Sorb NADES was calculated as 0.30 mmol/L in the sample diluted with 5 mL water. Consequently, the quantified value of the (*S*)-diastereomer was 36.7% higher than the intrinsic taste threshold (dose-over-threshold factor [58] of 1.6). The taste-modulating threshold is typically much lower than the intrinsic umami taste [29,59]. By diluting the *Maillard* model reaction of (***R***)**-2** and (***S***)**-2** in a glutamate solution, the taste-modulating impact on the overall taste can be increased. Therefore, the implemented model reaction in the Suc/Sorb NADES system can have a taste-enhancing impact in industry applications.

## 4. Conclusions

In conclusion, an inert Suc/Sorb NADES system comprising non-reductive sugar sucrose and polyol D-sorbitol was developed and characterized for the first time. NADES properties such as the supramolecular structure due to the formation of hydrogen bonds were successfully evaluated by NMR experiments. Reaction conditions were optimized and set at 100 °C for a maximum of three hours in a pH range between 5.7 and 9.0. A more extended heating period is conceivable with a simultaneous temperature reduction. However, the system’s stability should be checked individually for the selected applications and conditions before each reaction. As proof of principle, two *Maillard*-type model reactions to form taste-modulating 5′-GMP derivatives **1,** (***S***)**-2**, and (***R***)**-2** were successfully implemented in the new food-grade NADES system. All obtained yields (**1**: 95.7 µmol/g; sum of (***R***)**-2** and (***S***)**-2**: 161.8 µmol/mmol) were recalculated above the taste-modulating or intrinsic umami threshold concentrations and showed higher or comparable results to the literature data with respect to their formation in other NADES systems or aqueous buffered solutions. Due to the low toxicity and low reaction rate of the Suc/Sorb NADES system, it can be used as a solvent candidate in a wide variety of products and applications in the food industry, e.g., to process flavors or as a medium to generate taste-active or taste-modulating compounds in high concentrations. With further research and control of e.g., byproducts or microbial and sensory influences of Suc/Sorb NADES, such studies can provide the basis for a wide range of applications of NADES in the food industry. These studies are ongoing and will be published elsewhere.

## Figures and Tables

**Figure 1 foods-12-01877-f001:**
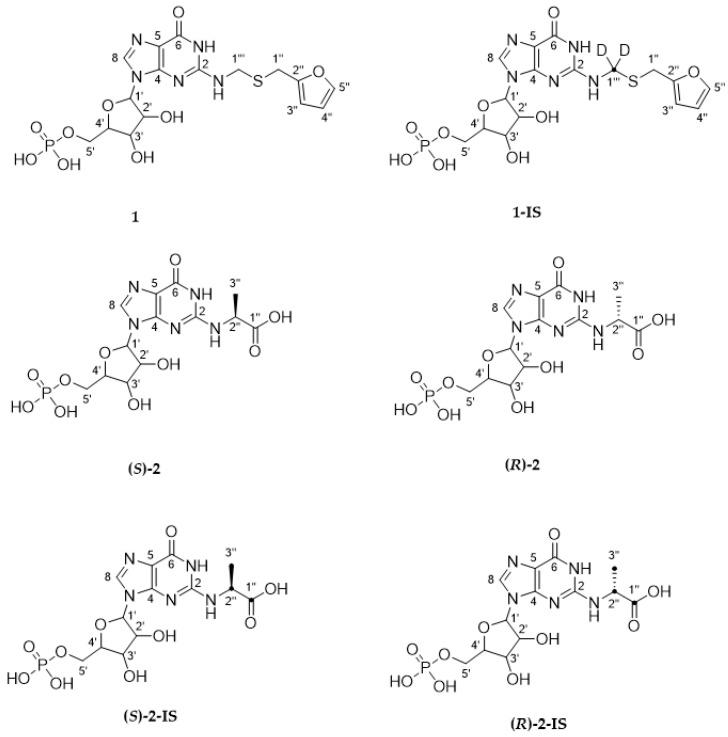
Chemical structures of *Maillard* model reaction products (**1**, (***S***)**-2** and (***R***)**-2**) produced in the Suc/Sorb NADES system, as well as their corresponding internal standards (**1-IS**, (***S***)**-2-IS**, and (***R***)**-2-IS**) used for quantification via UHPLC-MS/MS.

**Figure 2 foods-12-01877-f002:**
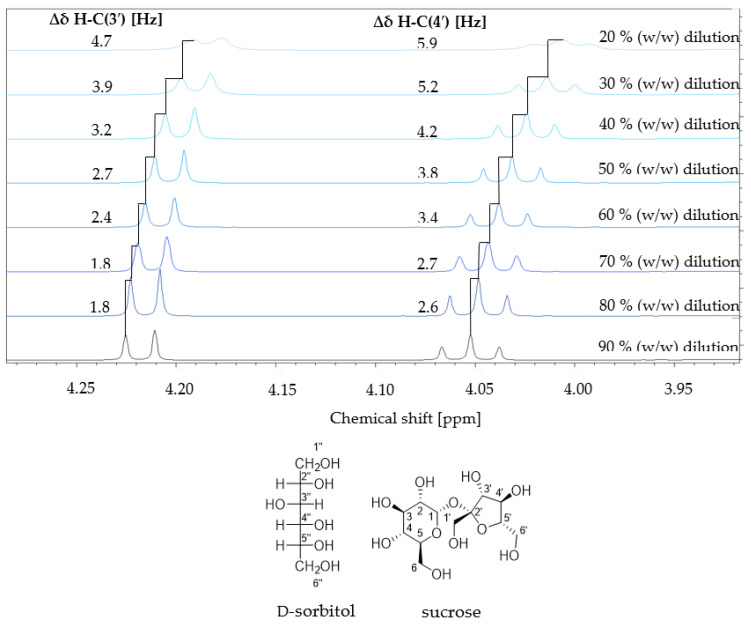
Excerpts of ^1^H-NMR titration spectra (600 MHz; D_2_O; 300 K) of a Suc/Sorb NADES system from 20–90% dilution (*w*/*w*) with D_2_O. The chemical shifts variations in Hz of two exemplary signals (H-C(3′) and H-C(4′) of sucrose) are depicted. Signals were referenced to TMSP.

**Figure 3 foods-12-01877-f003:**
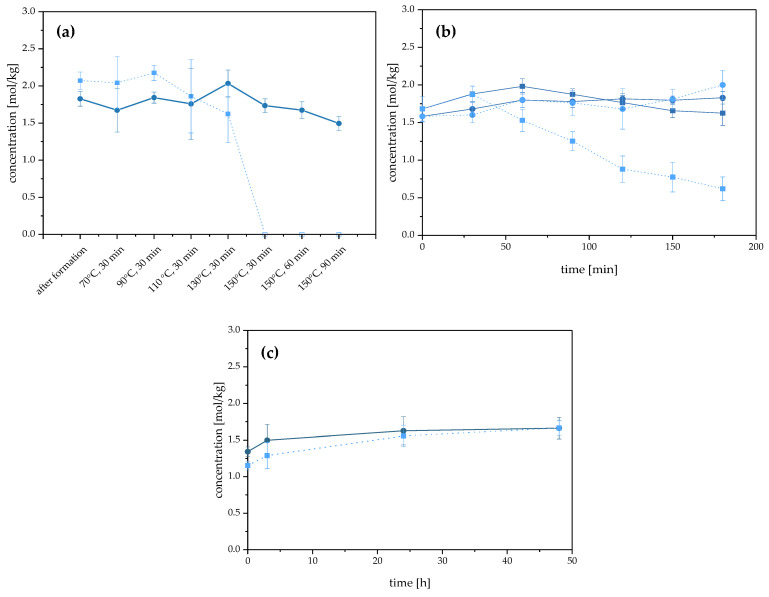
Thermal stability of Suc/Sorb NADES system by quantification of individual NADES ingredients (mol/kg) between 70 °C and 150 °C for 30 to 90 min (**a**) (
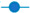
 D-sorbitol; 
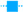
 sucrose); at 100 °C and 120 °C each for 180 min (**b**) (
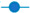
 D-sorbitol, 100 °C; 
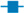
 sucrose, 100 °C; 
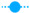
 D-sorbitol, 120 °C; 
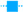
 sucrose, 120 °C); and at 40 °C for 48 h (**c**) (
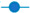
 D-sorbitol, 40 °C; 
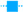
 sucrose). Measurement values are expressed as the mean of five replicates, with a minimum of two technical replicates and corresponding standard deviations.

**Figure 4 foods-12-01877-f004:**
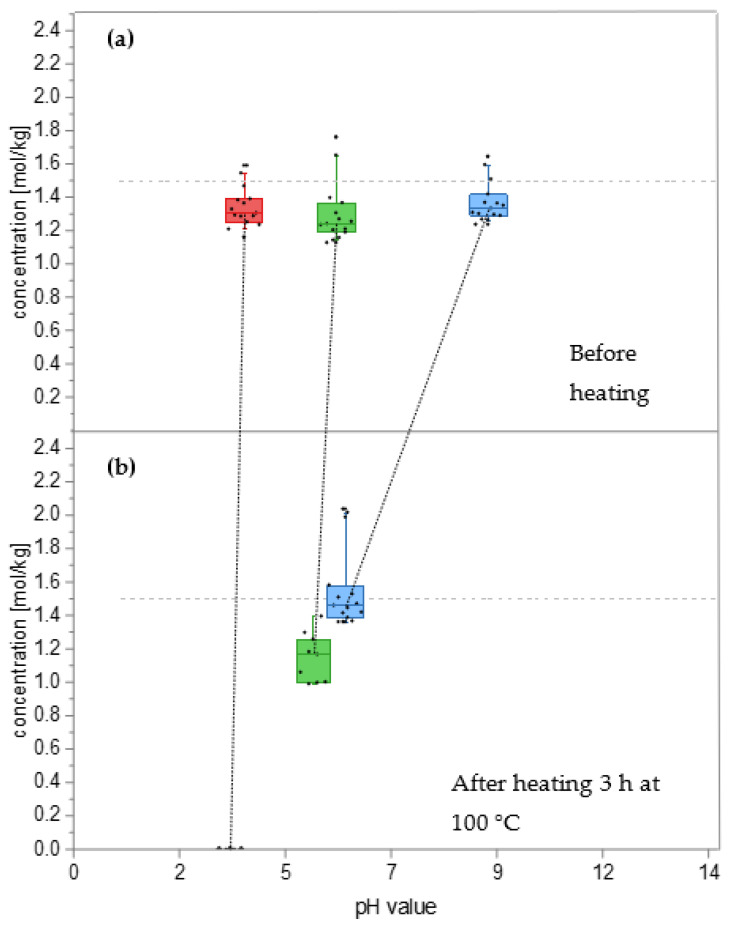
pH stability of Suc/Sorb NADES system by quantification of sucrose at (**a**) pH 3.7 (red), 5.7 (green), and 9.0 (blue); and (**b**) after heating at 100 °C for 180 min at pH 3.4 (red), 5.2 (green), and 5.9 (blue). Measurement values are expressed in boxplot diagrams. The control is for both NADES ingredients calculated at a concentration of 1.5 mol/kg.

**Figure 5 foods-12-01877-f005:**
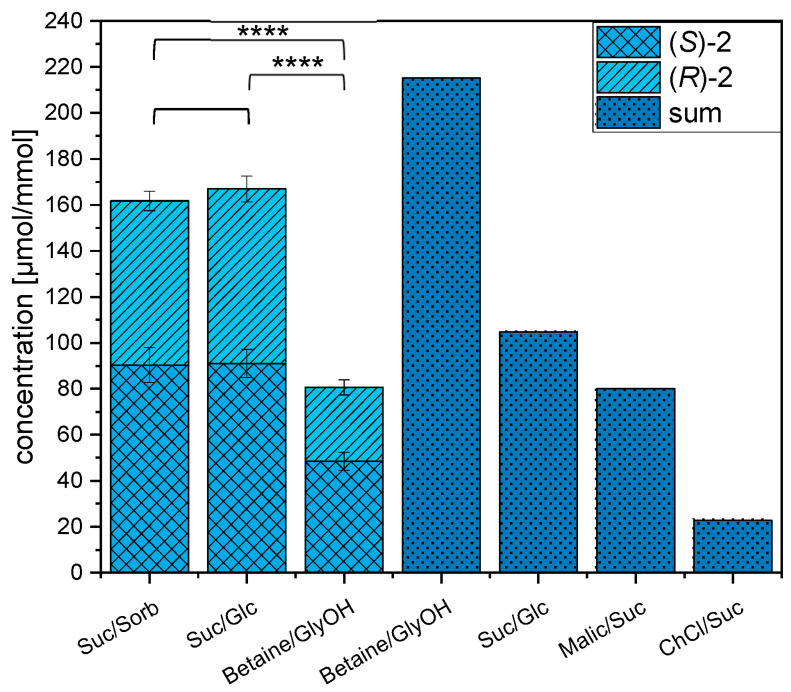
The yield of (***R***)**-2**, (***S***)**-2** (µmol/mmol) in Suc/Sorb, in sucrose/D-glucose (Suc/Glc), and betaine/glycerol (betaine/GlyOH) NADES systems, as well as from literature-adapted NADES systems as the sum of (***S***)**-2** and (***R***)**-2** (dark blue) betaine/GlyOH, Suc/Glc, malic acid/sucrose (Malic/Suc), and choline chloride/sucrose (ChCl/Suc) from Kranz and Hofmann [23]. Concentrations are reported in µmol per mmol D,L-glyceraldehyde. Measurement values are expressed as the mean of three replicates and corresponding standard deviations. Statistical differences were determined by means of the *t*-test for independent samples (*p* ≥ 0.05; **** *p* ≤ 0.0001).

## Data Availability

Data are contained within the article and Appendix A.

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
