# Peer review of "A New Inert Natural Deep Eutectic Solvent (NADES) as a Reaction Medium for Food-Grade *Maillard*-Type Model Reactions"

_foods, 2023, doi:10.3390/foods12091877_

Round 1

Reviewer 1 Report

Dear Authors

Comments are given in the attached file.

Author Response

Briefly include results and discussion in this section

Results and discussion included as recommended in the section lines 18-22

Few more experiments are required to use of NADES in food including microbial and sensory evaluation and thus this is not a proper conclusion

1) We agree with the reviewer that such expriments should be performed, these experiments are still in progress and will appear in our next manuskript later in this year. In particular, the results on degradation products and sensory analysis are beyond the scope of this manuscript and we have therefore decided to publish these data separately.

2) As recommended we changed some aspects in the conclusion line 568-590

Reviewer 2 Report

The application of natural deep eutectic solvents (NADES) ranges from organic chemistry to the agricultural sector and the food industry. In the food industry, the desired food quality can be achieved by the extraction of small molecules, macromolecules, and even heavy metals. The compound yield in Maillard-type model reactions can also be increased using NADES. To extend the so-called “kitchen-type chemistry” field, an inert, food-grade NADES system based on sucrose/dsorbitol was developed, characterized, and examined for its ability as a reaction medium by evaluating its temperature and pH stability. Reaction boundary conditions were determined at 100 °C for three hours and a pH range of 3.7–9.0. As proof of principle, two Maillard-type model reactions were implemented to generate the taste-modulating compounds N2(1carboxyethyl)guanosine 5monophosphate) and N2(furfurylthiomethyl)guanosine 5monophosphate. Since the yields of the two compounds are higher than their respective taste-modulating thresholds, the newly developed NADES is well suited for this type of “kitchen-type chemistry.”.

Overall, the manuscript is interesting and the topic about two Maillard-type model reactions were implemented. However, there still have some issues need to revise.

1.      “taste-modulating compounds N2(1carboxyethyl)guanosine 5monophosphate) and N2(furfurylthiomethyl)guanosine 5monophosphate” should be introduced.

2.      The Maillard reaction should be introduced in the introduced (Critical Reviews in Food Science and Nutrition. Doi: 10.1080/10408398.2022.2076064).

3.      NADES systems could conquer the yield limitations of Maillard model reactions and open a new field of “food-grade kitchen-type chemistry. In the meanwhile, the AGEs content should be analyzed (Food Chemistry, 417(2023):135861, Doi: 10.1016/j.foodchem.2023.135861).

4.      The reference should be updated.

5.      The grammar issues should be checked.

Author Response

  1. “taste-modulating compounds N2(1carboxyethyl)guanosine 5′monophosphate) and N2(furfurylthiomethyl)guanosine 5′monophosphate” should be introduced. Compund introduced as recommended in the lines 63-71.
  2. The Maillard reaction should be introduced in the introduced (Critical Reviews in Food Science and Nutrition. Doi: 10.1080/10408398.2022.2076064). The content of the paper suggested by reviewer 2 has nothing to do with nothing to do with the work presented by us. AGEs are products formed by carbohydrate degradation produchts an e.g. the ε-amino moiety of proteins. There are no proteins involved in our model-reactions. And the consumption of sugar and sugar degradation product in generall will lead to AGEs in the human body during life time - so, at the end from our point of view it makes no sence to discuss this fact within this manuscript.

  3. Paper deals with milk and diary products with high concentration of proteins and has nothing to do with the work presented in our manuscript.
  4. Due to the inconsistent subject matter of the literature suggested by the reviewer, we would like to refrain from citing them.
  5. The manuscript was carefully revised by a professional proof reading service (native speaker)  see pdf attached.
